

# A consistent regional dataset of dissolved oxygen in the Western Mediterranean Sea (2004-2023): O2WMED

Malek Belgacem[1], Katrin Schroeder[1], Siv K. Lauvset[2], Marta Álvarez[3], Jacopo Chiggiato[1], Mireno Borghini[4], Carolina Cantoni[5], Tiziana Ciuffardi[6], Stefania Sparnocchia[5]

[1] CNR-ISMAR, Arsenale Tesa 104, Castello 2737/F, 30122 Venice, Italy
[2] NORCE Norwegian Research Centre, Bjerknes Centre for Climate Research, Bergen, Norway
[3] Instituto Español de Oceanografía, IEO-CSIC, A Coruña, Spain
[4] CNR-ISMAR, Via Santa Teresa, Pozzuolo di Lerici, 19032 La Spezia, Italy
[5] CNR-ISMAR, Area Science Park, Basovizza, 34149 Trieste, Italy
[6] Department of Sustainability, St Teresa Marine Environment Research Centre, ENEA, Pozzuolo di Lerici, 19032 La Spezia, Italy

*Correspondence to:* Malek Belgacem (malek.belgacem@ve.ismar.cnr.it)

**Abstract.** A new dataset from oceanographic cruises in the Western Mediterranean Sea (WMED) was compiled to integrate the previously published regional data product CNR-DIN-WMED about dissolved inorganic nutrients (https://doi.org/10.1594/PANGAEA.904172, Belgacem et al., 2019, 2020). The Mediterranean region is experiencing rapid changes, necessitating high-quality and reliable datasets. However, the scarcity of the in-situ observations hinders the understanding of these changes and their impact on biogeochemical cycles. Dissolved oxygen is a vital component of marine ecosystems and plays a fundamental role in governing nutrient and carbon cycles, underscoring the need for accurate and reliable data. To address this, a high resolution, regional-scale data product was developed to understand decadal variability and spatial/temporal patterns of the ventilation process in the WMED. This study presents an extensive collection of unpublished dissolved oxygen data from continuous sensors collected between 2004 and 2023, along with a description of the quality assurance procedures. The quality assurance process involves calibration of CTD measurements against Winkler analyses and the comparison of deep observations with reference datasets, using the crossover analysis. The resulting data product O2WMED can be used as reference for assessing oxygen sensors mounted on biogeochemical Argo (BGC-Argo) floats or Gliders and for regional model validation.

**Data coverage**

Coverage: 44∘ N–35∘ S, 6∘ W–14°E
Location name: western Mediterranean Sea
Date/time start: October 2004
Date/time end: April 2023

## 1   Introduction

Oxygen plays a crucial role as a fundamental oceanic variable. The ocean produces about 50% of the earth's oxygen, which is crucial for the atmospheric oxygen inventory (Grégoire et al. 2023). The ocean's oxygen levels are highly susceptible to changes, through



Photosynthesis occurs in the surface layer, where oxygen is produced by primary producers, leading to high surface productivity and cause the sinking of organic matter which stimulate oxygen consumption in the deep sea, leading to the creation of the Oxygen Minimum zones (OMZs). The OMZs have become more frequent in the last decade.

Increased degradation of organic matter in the deep ocean intensifies dissolved oxygen consumption. Warming conditions contribute to the heightened stratification periods and an intensification of the pycnocline, impacting biological activity and thus dissolved oxygen consumption. Additionally, factors such as denitrification and the extension of the OMZs can influence the nitrogen: phosphorus (N:P) ratio and the levels of primary productivity. Warming seawater, increased stratification and higher $CO_2$ levels have reshaped community distribution and ecosystem composition, resulting in decreased dissolved oxygen levels (Grégoire et al., 2023).

The decline in oxygen can be attributed to lower solubility rates due to warming and increased water column stratification, reducing the downward diffusion and mixing of well-oxygenated surface waters towards deeper layers. Such changes in oxygen concentration can trigger variations in the biological loop and the distribution of biogeochemical tracers, which have significant implications for marine ecosystems, particularly in vulnerable hotspots.

The Mediterranean Sea has been notably affected by these changes, especially in the past decade, characterized by frequent marine heat waves episodes (Marullo et al., 2023; Martinez et al., 2023; Pastor and Khodayar et al., 2023). Rising temperatures can disturb the distribution and availability of dissolved oxygen in seawater (Reale et al., 2022; Alvaréz et al. 2023). The region's enclosed nature and its peculiar thermohaline circulation further complicates the understanding of oxygen dynamics (Powley et al., 2016). Observational efforts, such as the Medar/Medatlas project (Fichaut et al., 2003), the MED-SHIP transects ([https://www.go-ship.org/](https://www.go-ship.org/)) (Schroeder et al., 2015), have provided valuable insights, but uncertainties remain regarding the long-term impacts of deoxygenation and acidification on Mediterranean marine ecosystems (Coppola et al., 2018).

To address these pressing issues, a compilation of oxygen observations collected by the Italian National Research Council (CNR), between 2004 and 2023 in the Western Mediterranean Sea (WMED) is documented. This effort aims to provide reliable measurements of dissolved oxygen, thereby enhancing our understanding of biogeochemical cycling and ventilation in the region. The purpose of this paper is to describe the oxygen data collected by the CNR and report the quality control procedure to justify the recommended corrections applied to the oxygen data.

## 2   Dissolved oxygen data collection

### 2.1   The CNR data collection

The **O**xygen in the **WMED** (O2WMED) dataset contains 1,382 CTD oxygen profile. In Figure 1, the spatial distribution of CTD profiles is depicted, highlighting the extensive coverage across the Northern Western Mediterranean (WMED) and key hydrographic transects.

Measures are concentrated in the eastern part of the WMED: the subregions of the Ligurian sea, Tyrrhenian and along the Tunisia-Sicily-Sardinia area. Spanning two decades from 2004 to 2023, the dataset exhibits robust temporal coverage, particularly between 2004 and 2015 (see Fig.2a). In particular, the years 2005, 2006, 2010, and 2012 stand out with the highest number of CTD stations, coinciding with years that included monthly surveys,





76    indicating a more frequent repeat frequency (see Fig. 2). While reasonable temporal coverage is observed between

77    2004 and 2015 (except for 2014), the availability of stations diminishes between 2016 and 2023.

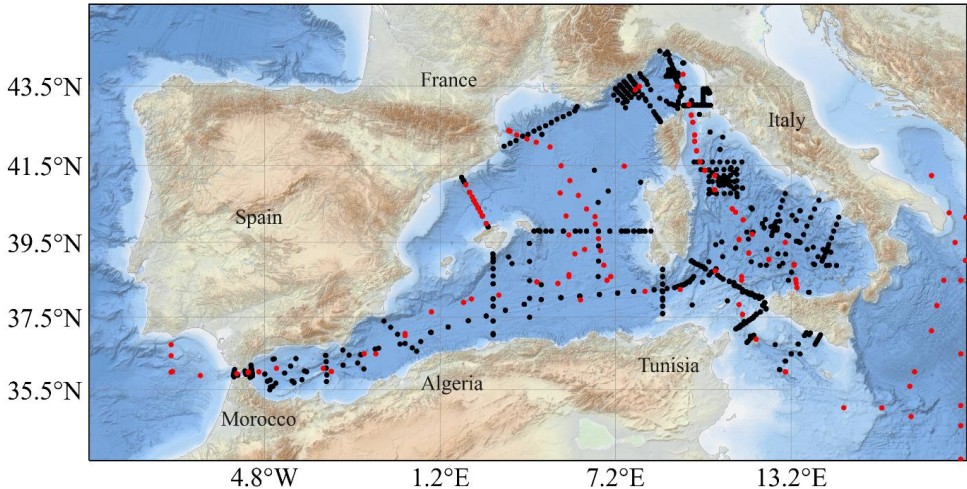

78

**Figure 1. Spatial distribution of cruise stations with CTD oxygen data (black dots) in the O2WMED CNR dataset across the WMED. The red markers indicate stations from selected reference cruises.**

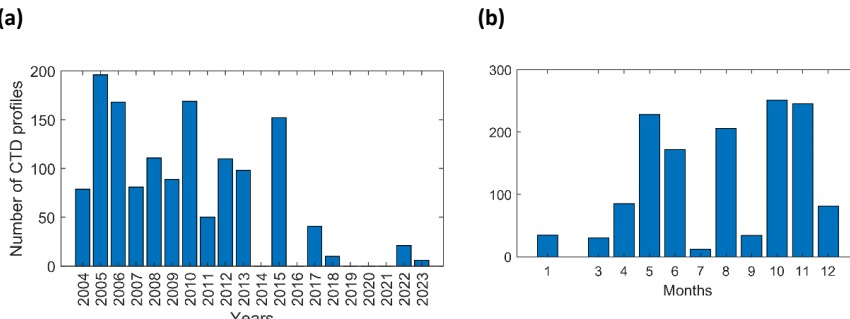

**Figure 2. Temporal distribution of CTD profiles with oxygen in the O2WMED CNR dataset: A. annual distribution and B. monthly distribution.**

Data included in the dataset (Table 1) were routinely calibrated against Winkler measurements following Grasshoff
et al. (1983) and Langdon (2010). Discrete samples were used to calibrate the CTD sensor to correct any potential
bias and adjust drift in the SBE 43 oxygen sensor following Janzen et al. (2007) and Uchida et al. (2010). Details
regarding the post-calibration against Winkler observation can be found in the supplementary materials (Table
S1).











**Table 1. Cruise summary table listed with number of stations. Refer to Belgacem et al. (2020) and Ribotti et al. (2022) for cruise metadata.**

| Cruise ID no. | Common name | EXPOCODE | Date Start/End | CTD profiles | Maximum bottom depth (m) |
|---|---|---|---|---|---|
| 2 | MEDGOOS9 | 48UR20041006 | 6 - 25 OCT 2004 | 79 | 3668 |
| 3 | MEDOCC05/ MFSTEP2 | 48UR20050412 | 24 APR - 16 MAY 2005 | 160 | 3657 |
| 5 | MEDGOOS11 | 48UR20051116 | 16 NOV - 3 DEC 2005 | 36 | 3494 |
| 6 | MEDOCC06 | 48UR20060608 | 8 JUN - 3 JUL 2006 | 127 | 2882 |
| 8 | MEDGOOS13/MEDBIO06 | 48UR20060928 | 28 SEP - 8 NOV 2006 | 41 | 4137 |
| 9 | MEDOCC07 | 48UR20071005 | 5 - 29 OCT 2007 | 81 | 3497 |
| 10 | SESAMEIt4 KM3 or SESAME_KM3 | 48UR20080318 | 18 MAR - 7 APR 2008 | 27 | 3510 |
| 11 | SESAMEIT5 (Sesame KM3 September 2008) | 48UR20080905 | 5 - 16 SEP 2008 | 24 | 3450 |
| 12 | MEDCO08 | 48UR20081103 | 3 - 24 NOV 2008 | 60 | 3443 |
| 13 | TYRRMOUNTS | 48UR20090508 | 8 MAY - 3 JUN 2009 | 89 | 3509 |
| 14 | BIOFUN010 | 48UR20100430 | 30 APR - 17 MAY 2010 | 29 | 3541 |
| 15 | VENUS1 | 48UR20100731 | 31 JUL - 25 AUG 2010 | 116 | 3649 |
| 16 | BONSIC2010 | 48UR20101123 | 23 NOV - 9 DEC 2010 | 24 | 3539 |
| 17 | EUROFLEET11 | 48UR20110421 | 21 APR - 8 MAY 2011 | 32 | 3541 |
| 18 | BONIFACIO2011 | 48UR20111109 | 9 - 23 NOV 2011 | 18 | 3542 |
| 20 | ICHNUSSA12 | 48UR20120111 | 11 - 27 JAN 2012 | 35 | 3552 |
| 21 | EUROFLEET2012 | 48UR20121108 | 8 - 26 NOV 2012 | 75 | 3554 |
| 211 | VENUS 2 | 48UR20130604 | 4 - 25 JUN 2013 | 59 | 3539 |
| 22 | ICHNUSSA13 | 48UR20131015 | 15 - 29 OCT 2013 | 40 | 3542 |
| 222 | ICHNUSSA15 | 48QL20151123 | 23 NOV - 14 DEC 2015 | 62 | 3633 |
| 23 | OCEANCERTAIN15 | 48QL20150804 | 4 - 18 AUG 2015 | 90 | 3514 |
| 24 | ICHNUSSA17/INFRAOCE17 | 48QL20171023 | 23 OCT- 28 NOV  2017 | 41 | 3537 |
| 25 | ICHNUSSA/JERICO18 | 48DP20180918 | 18-25 SEP 2018 | 10 | 525 |
| 27 | JERICO-II-2022 | 48DP20221015 | 15- 25 OCT 2022 | 21 | 1004 |
| 28 | JERICO-III-EurogoShip-2023 | 48DP20230324 | 24 MAR - 09 APR 2023 | 6 | 909 |


The vertical distribution of dissolved Oxygen in the WMED exhibits a distinct pattern across different depth layers as illustrated in the vertical profiles shown in Fig.3(a). These profiles, which span ten subregions, provide a comprehensive view of spatial coverage, considering the depth component in conjunction with the reference dataset.

Broadly speaking, profiles reveal a general trend of high concentrations in both the upper and bottom layers, with an intermediate oxygen minimum layer, typically observed between 400 and 600 meters (Mavropoulou et al., 2020). At greater depths, subregions exhibit distinct behaviors. The dissimilarities in the vertical distribution highlight regional variations, influenced by factors such as circulation patterns, biological activity, and the unique physical characteristics of each region. This is particularly true for the Mediterranean, characterized by a complex circulation pattern involving inflows from the Atlantic Ocean, surface currents, and deep-water formation. Circulation patterns play a crucial role in transporting oxygen-rich or oxygen-poor water masses to different regions (Mavropoulou et al., 2020). For instance, in Figure 3a the DF2-Gulf of Lion, oxygen-rich waters reach the deep WMED by means of winter deep convection.



**(a)**

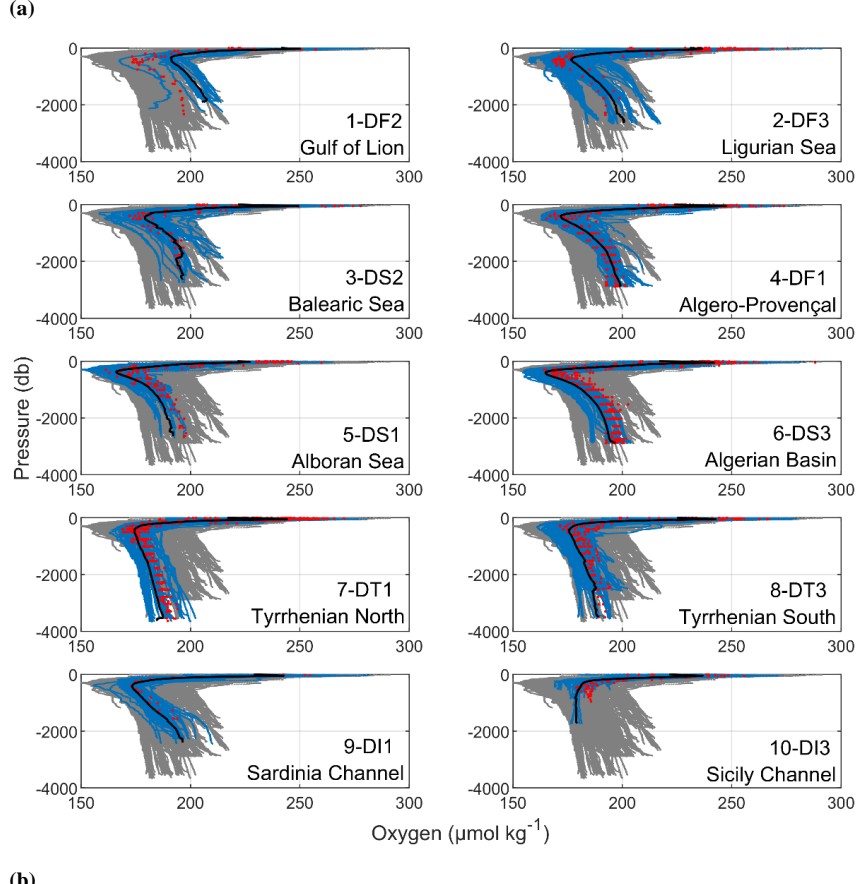

**(b)**

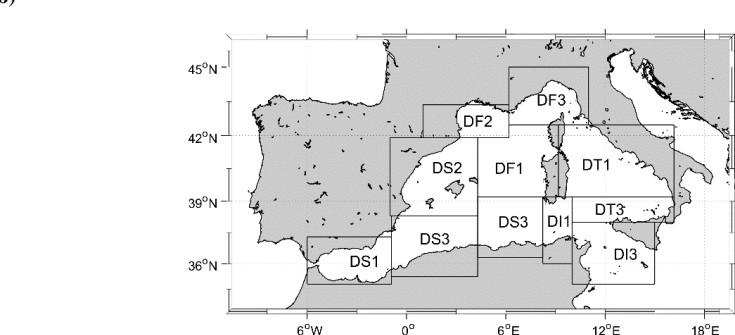

**Figure 3. (a) Vertical profiles of oxygen distribution for the entire WMED in grey, from the original dataset after the**
**initial quality control (1st QC). The blue represents the vertical distribution within each subregion (refer to Table S1).**
**The black lines denote the mean profile of each subregion over the entire study period. The red lines represent the**
**reference data for each region. (b) Geographical map of the WMED indicating the geographical limits from**
**MEDAR/Medatlas sub-regions (defined in Table S1). Adapted from Manca et al. (2004).**
**2.2   Reference data**
The previously described CNR collection data is compared to deep water data collected in the same area (details
are in section 3.2).



A total of six cruises have been identified (see Table 2) with documented high-quality dataset (Figure 4), collected
in the Mediterranean Sea through international projects. Defined as "reference cruises," they adhere to the
recommendations of the World Ocean Circulation Experiment (WOCE) and the Global Ocean Ship-Based
Hydrographic Investigations Program (GO-SHIP) protocols (Langdon 2010). Among these, cruises
06MT20011018 and 06MT20110405 are significant surveys, contributing to the GLODAPv2 dataset (Olsen et al.,
2016), which facilitate comprehensive mapping of biogeochemical parameters.
During cruise both cruises, quality control procedures were applied to data, and minimal corrections to oxygen
measurements were made, ensuring excellent data quality. Additional information regarding these cruises can be
found in the work of Tanhua et al. (2013a) and Hainbucher (2012). For further details regarding these references,
one can refer to the adjustment table available at  https://glodap.info/ (last access: August 2024) and the work by
Olsen et al. (2020).
Similarly cruises 48UR20070528 (TRANSMED II) and 29AH20140426 (HOTMIX) which are related to
CARIMED (CARbon, tracer and ancillary data In the MEDsea) that aims to be an internally consistent database
containing inorganic carbon data relevant for this basin (Álvarez et al., in preparation); which means that this
cruise underwent rigorous quality control processes
The TALPpro cruises conducted in 2016 (Tanhua ,2019a, 2019b; Jullion, 2016) and in 2022 (Schroeder, 2022) are
associated with the MedSHIP program, which follows the guidelines established by the international GO-SHIP
initiative. This program is dedicated to high-quality data collection and analysis to evaluate the impacts of climate
on marine environments (Schroeder et al., 2015, 2024).
Following the standards and charts of these programs, the quality of measurements obtained during these cruises
has been ensured, demonstrating both precision and reliability, and thus used as reference cruises in the secondary
quality control procedure described below in section 3.2.
**Table 2. Overview of reference cruises utilized in the secondary quality control process with their Expocode and**
**Identification number (ID). The data spans from 2001 to 2022.**

| ID | Common name | EXPOCODE | Date starts and end | Stations | Source | Chief scientist(s) |
|---|---|---|---|---|---|---|
| 6 | M51/2 | 06MT20011018 | 18 Oct–11 Nov 2001 | 6 | GLODAPv2 | Wolfgang Roether |
| 22 | TRANSMED_LE GII | 48UR20070528 | 28 May–12 Jun 2007 | 4 | CARIMED | Maurizo Azzaro |
| 64 | M84/3 | 06MT20110405 | 5–28 Apr 2011 | 20 | GLODAPv2 | Toste Tanhua |
| 17 | HOTMIX | 29AH20140426 | 26 Apr–31 May 2014 | 18 | CARIMED | Javier Aristegui |
| 27 | TAlPro-2016 | 29AJ20160818 | 18–28 Aug 2016 | 42 | MedSHIP programme | Loïc Jullion, Katrin Schroeder |
| 28 | TAlPro-2022 | 11BG20220517 | 17-26 May 2022 | 24 | MedSHIP/ MedSHIP programme | Katrin Schroeder |




(a)                                                          (b)

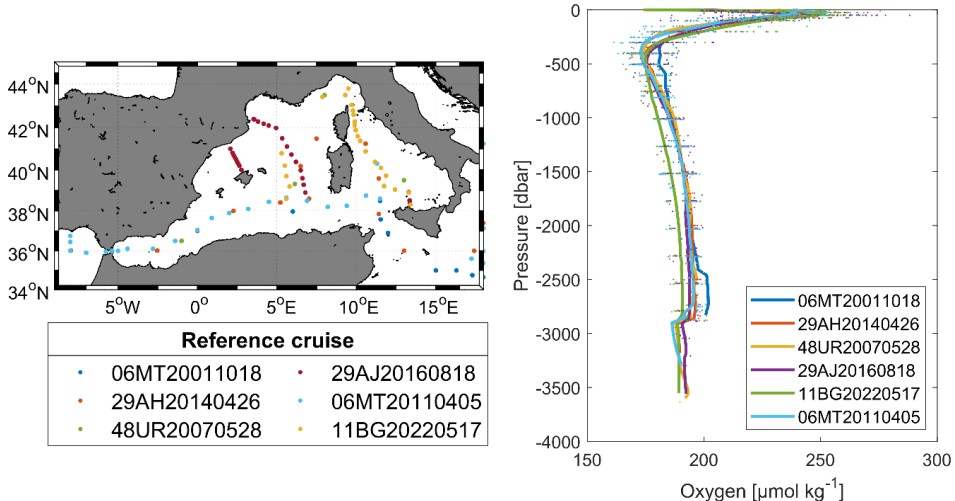

**Figure 4. Reference cruises: (a) Map with stations. (b) Dissolved oxygen data from Winkler measurements and corresponding mean profile from the reference cruises.**

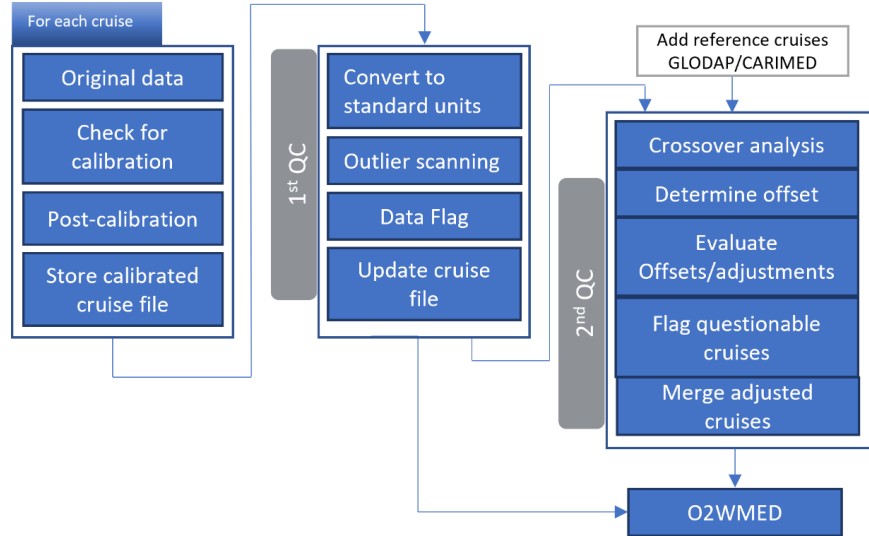

**Figure 5. Flowchart illustrating the sequential steps from calibration through primary quality control (1stQC) to secondary quality control; refer to the text for comprehensive details.**

## 3 Quality assurance methods

### 3.1 Primary quality control of O2 CTD data

Following Figure 5 chart, each cruise variable was scanned for any spike before calibration. Then values were converted to standard units when needed. Dissolved oxygen was converted from milliliters per liter (ml L$^{-1}$) to micromoles per kilogram (µmol kg$^{-1}$) using potential density and the conversion factor 44.66 to ensure uniformity.
For a thorough data integrity, each parameter was assigned a data quality flag. Initially set to 2 for acceptable
values, flag 3 for questionable values and to 9 for data following WOCE.
The primary quality control (1$^{st}$ QC) procedure involved the identification of outlier profiles and/ or data points in
each cruise. Outliers were flagged, indicating the quality of each value (refer to Table 3 in Belgacem et al., 2020).
Flagging was specific to the precision of each parameter for each cruise. Property-property plots were then
examined for each region, and values identified as outliers in more scatter plots were flagged as questionable (see
Fig. 6).
A scatter plot depicting the oxygen distribution provided an overview, with values flagged as 3 for questionable
values or 2 for accepted values. Approximately 0.19% of CTD oxygen data were considered outliers and flagged
as 3. The 1$^{st}$ QC can be subjective, as it relies on the expertise of the individual inspecting the data.
The coefficient of variation of Oxygen profiles (CV, defined as standard deviation over mean) for each layer
(surface: 0-250 db; intermediate: 250-1000 db; deep: below 1000 db) has been considered. CV in the surface layer
(0-250 db, CV = 11.7%) were relatively high due large frequency variability (caused by air-sea interaction), at
intermediate levels (250-1000db, CV = 4.5%), and deep layer (below 1000 db, CV =4.4%), these variabilities are
reduced.

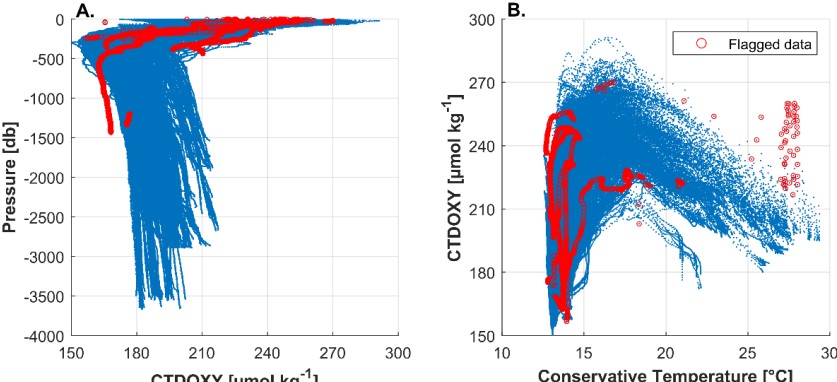


**Figure 6. Summary scatter plots illustrating (A) pressure vs. Oxygen and (B) Oxygen vs. potential temperature while**
**red circles indicate values flagged as questionable (Flag 3).**
To evaluate the precision of each cruise data, adjacent profiles were compared. Standard deviations and averages
were calculated for deep layers (depths greater than 1000 dbar, as shown in Table 3) to sidestep any variability
associated with atmospheric forcing or mesoscale patterns.
Data with poor precision are expected to show large standard deviations, indicating significant discrepancies from
nearby measurements.
Additionally, this analysis provides an overview of oxygen content in deep layers and the spatial extent of
measurements for each survey.  Following the subdivision of the WMED proposed by Manca et al. (2004) (see
Fig. 3b and Table S1), comparison of regional averages allowed the identification of potentially suspect cruises.



The standard deviation between cruises in deep layers varied between 0.5 and 7.6 µmol kg$^{-1}$. Overall, profiles
collected in close proximity exhibit similar variability.
The lowest standard deviations were observed in data collected from the Sicily Channel (DI3) subregions during
cruises no. 10, no. 11, and no. 211, indicating high data quality for these surveys.
Fifteen surveys were conducted in Sardinia Channel (DI1), all displaying similar variability. Standard deviation
among cruises in this region ranged between 4 to 6.5 µmol kg$^{-1}$, demonstrating good agreement amid
measurements.
A similar pattern was observed in the Tyrrhenian South region (DT3), which was samples by 22 cruises.
The lowest average was recorded during cruise #2, which was lower than neighboring cruises, while cruise #24
exhibited the highest average; however, the standard deviation did not show significant variation.
In the Tyrrhenian North region (DT1), eighteen cruises were conducted, with standard deviations ranging from 1.4
to 5.5. Notably, cruise no. 3 displayed a larger standard deviation of 7.5 µmol kg$^{-1}$.
In the Algerian basin (DS3), ten cruises were conducted, yielding standard deviations between 3.3 and 4.4 µmol
kg$^{-1}$, similar to the findings in the Alboran Sea (DS1).
In the Algéro-Provençal region (DF1), we find data from eleven cruises, with the highest standard deviations
recorded during cruises no. 3 and no. 24.
In the Balearic Sea (DS2), six cruises sampled the deep layer, all exhibiting low standard deviations, except for
cruises #3 and #6, which had values of 6.1 and 7.6 µmol kg$^{-1}$ respectively.
In the Ligurian Sea (DF3), five cruises where conducted, with cruise no.3 displaying anomalous behavior
characterized by a high standard deviation and average compared to neighboring cruises, indicating that the data
from cruise #3 were higher the nearby measurements.
High standard deviations suggest extensive spatial coverage, as seen in cruises no. 3, no. 6, no.12 and no.15, while
low standard deviation, such as in cruise no. 11 (DI3-Sicily Channel), no. 17, and no. 8 (predominantly in DT1-
Tyrrhenian North and DT3-Tyrrhenian South), suggests smaller spatial coverage.
The condition suggested by Olsen et al. (2016), which suggests that large standard deviation coupled with narrow
spatial coverage implies imprecise data, was not found in our dataset.
An examination of data spread across various regions revealed that some cruises exhibited large standard
deviations compared to nearby profiles. For instance, cruise no.3 (Tyrrhenian North, Balearic Sea and Ligurian
Sea) had standard deviations exceeding 6.1 µmol kg$^{-1}$, suggesting that this cruise was less precise than the others
conducted in the same region.
Comparing deep profiles provided evidence regarding data precision; uncertainties in measurements may
complicate decisions about the adjustments proposed by the 2$^{nd}$ QC phase.
Cruises no. 25, no. 27, no. 28 did not include data below 1000 dbar; however, these datasets underwent quality
checks and are included in the final dataset.

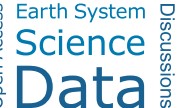

**Table 3. Average and standard deviation (STD) of CTD dissolved oxygen by cruise and for each region deeper than**
**1000 db.**

| Cruise ID | EXPOCODE/ region | Regional avg CTDOXY (µmol kg⁻¹) | STD CTDOXY (µmol kg⁻¹) |
|---|---|---|---|
| 2 | 48UR20041006 / | | **4.193** |
| | DS2-Balearic Sea | 182.48 | 3.27 |
| | DS1-Alboran Sea | 181.71 | 3.59 |
| | DS3-Algerian Basin | 183.59 | 3.32 |
| | DT1-Tyrrhenian North | 177.49 | 3.21 |
| | DT3-Tyrrhenian South | 177.57 | 3.09 |
| | DI1-Sardinia Channel | 181.02 | 4.17 |
| 3 | 48UR20050412/ | | **9.538** |
| | DF2-Gulf of Lion | 199.65 | 5.64 |
| | DF3-Ligurian Sea | 183.42 | 7.46 |
| | DS2-Balearic Sea | 200.67 | 6.15 |
| | DF1-Algero-Provençal | 197.87 | 5.76 |
| | DS3-Algerian Basin | 196.17 | 4.44 |
| | DT1-Tyrrhenian North | 191.03 | 4.67 |
| | DT3-Tyrrhenian South | 189.31 | 4.13 |
| | DI1-Sardinia Channel | 191.27 | 4.68 |
| 5 | 48UR20051116/ | | **3.542** |
| | DT1-Tyrrhenian North | 191.73 | 2.79 |
| | DT3-Tyrrhenian South | 190.22 | 3.90 |
| 6 | 48UR20060608/ | | **9.76** |
| | DF2-Gulf of Lion | 207.71 | 3.09 |
| | DF3-Ligurian Sea | 207.72 | 3.85 |
| | DS2-Balearic Sea | 195.00 | 7.60 |
| | DF1-Algero-Provençal | 194.82 | 4.95 |
| | DS3-Algerian Basin | 187.81 | 3.75 |
| | DT3-Tyrrhenian South | 185.06 | 3.49 |
| | DI1-Sardinia Channel | 188.80 | 4.21 |
| 8 | 48UR20060928/ | | **2.812** |
| | DT1-Tyrrhenian North | 177.87 | 2.81 |
| | DT3-Tyrrhenian South | 178.15 | 2.80 |
| 9 | 48UR20071005/ | | **5.77** |
| | DF2-Gulf of Lion | 185.57 | 1.38 |
| | DF3-Ligurian Sea | 190.87 | 3.37 |
| | DS2-Balearic Sea | 193.20 | 1.61 |
| | DF1-Algero-Provençal | 189.95 | 4.01 |
| | DS3-Algerian Basin | 189.55 | 4.16 |
| | DT1-Tyrrhenian North | 179.48 | 1.41 |
| | DT3-Tyrrhenian South | 180.09 | 2.18 |
| | DI1-Sardinia Channel | 183.39 | 4.15 |
| 10 | 48UR20080318/ | | **5.079** |
| | DT3-Tyrrhenian South | 187.15 | 4.72 |
| | DI3-Sicily Strait | 180.18 | 0.72 |
| 11 | 48UR20080905/ | | **0.610** |
| | DI3-Sicily Channel | 177.74 | 0.61 |
| 12 | 48UR20081103/ | | **4.81** |
| | DF1-Algero-Provençal | 192.51 | 4.04 |
| | DS1-Alboran Sea | 190.61 | 3.17 |
| | DS3-Algerian Basin | 193.71 | 3.81 |
| | DT3-Tyrrhenian South | 184.63 | 3.14 |
| | DI1-Sardinia Channel | 188.58 | 4.53 |
| 13 | 48UR20090508/ | | **3.044** |
| | DT1-Tyrrhenian North | 180.42 | 1.59 |
| | DT3-Tyrrhenian South | 181.78 | 2.83 |
| | DI1-Sardinia Channel | 187.67 | 4.75 |
| 14 | 48UR20100430/ | | **5.44** |
| | DS2-Balearic Sea | 194.39 | 3.15 |
| | DF1-Algero-Provençal | 192.67 | 4.69 |
| | DS3-Algerian Basin | 193.28 | 3.55 |
| | DT1-Tyrrhenian North | 184.26 | 2.23 |
| | DT3-Tyrrhenian South | 183.86 | 2.02 |
| | DI1-Sardinia Channel | 186.52 | 5.23 |
| 15 | 48UR20100731/ | | **6.21** |
| | DS1-Alboran Sea | 186.58 | 3.68 |
| | DS3-Algerian Basin | 189.94 | 3.34 |
| | DT1-Tyrrhenian North | 178.03 | 2.22 |
| | DT3-Tyrrhenian South | 178.14 | 3.27 |
| | DI1-Sardinia Channel | 186.60 | 5.18 |
| 16 | 48UR20101123/ | | **3.96** |
| | DT1-Tyrrhenian North | 186.29 | 3.86 |



| | | | |
|---|---|---|---|
| | DT3-Tyrrhenian South | 181.04 | 1.68 |
| 17 | 48UR20110421/ | | **2.47** |
| | DT1-Tyrrhenian North | 180.0 | 1.86 |
| | DT3-Tyrrhenian South | 179.4 | 2.80 |
| 18 | 48UR20111109/ | | **5.62** |
| | DF1-Algero-Provençal | 191.20 | 3.92 |
| | DT1-Tyrrhenian North | 182.00 | 2.22 |
| | DT3-Tyrrhenian South | 182.79 | 3.15 |
| | DI1-Sardinia Channel | 185.19 | 5.43 |
| 20 | 48UR20120111// | | **5.29** |
| | DF1-Algero-Provençal | 192.90 | 4.35 |
| | DT1-Tyrrhenian North | 184.69 | 3.06 |
| | DT3-Tyrrhenian South | 184.17 | 2.88 |
| | DI1-Sardinia Channel | 186.06 | 5.60 |
| 21 | 48UR20121108/ | | **5.73** |
| | DF3-Ligurian Sea | 197.73 | 5.27 |
| | DT1-Tyrrhenian North | 190.86 | 3.05 |
| | DT3-Tyrrhenian South | 189.89 | 2.85 |
| | DI1-Sardinia Channel | 196.25 | 5.42 |
| 211 | 48UR20130604/ | | **6.086** |
| | DF1-Algero-Provençal | 192.907 | 4.337 |
| | DS3-Algerian Basin | 191.516 | 4.118 |
| | DT1-Tyrrhenian North | 183.228 | 3.139 |
| | DT3-Tyrrhenian South | 182.148 | 3.147 |
| | DI1-Sardinia Channel | 186.594 | 5.264 |
| | DI3-Sicily Channel | 178.630 | 0.531 |
| 22 | 48UR20131015/ | | **6.03** |
| | DF1-Algero-Provençal | 195.96 | 4.90 |
| | DS3-Algerian Basin | 196.54 | 4.24 |
| | DT1-Tyrrhenian North | 188.79 | 4.38 |
| | DT3-Tyrrhenian South | 187.66 | 3.92 |
| | DI1-Sardinia Channel | 190.87 | 6.03 |
| 222 | 48QL20151123/ | | **4.27** |
| | DT1-Tyrrhenian North | 183.17 | 3.84 |
| | DT3-Tyrrhenian South | 181.97 | 3.17 |
| | DI1-Sardinia Channel | 185.91 | 5.48 |
| 23 | 48QL20150804/ | | **5.79** |
| | DF3-Ligurian Sea | 190.75 | 4.91 |
| | DS2-Balearic Sea | 191.13 | 3.21 |
| | DF1-Algero-Provençal | 190.32 | 4.43 |
| | DS3-Algerian Basin | 191.41 | 3.85 |
| | DT1-Tyrrhenian North | 184.60 | 3.60 |
| | DT3-Tyrrhenian South | 180.45 | 3.24 |
| 24 | 48QL20171023/ | | **7.195** |
| | DF1-Algero-Provençal | 200.42 | 6.49 |
| | DT1-Tyrrhenian North | 193.62 | 5.53 |
| | DT3-Tyrrhenian South | 190.62 | 4.26 |
| | DI1-Sardinia Channel | 201.79 | 6.59 |
| 25[a] | 48DP20180918/ | | |
| | DI3-Sicily Channel | - | - |
| 27[a] | 48DP20221015/ | | |
| | DI3-Sicily Channel | - | - |
| 28[a] | 48DP20230324/ | | |
| | DI3-Sicily Channel | - | - |

[a] Cruises not included in the second QC. In bold: the overall standard deviation by cruise; in normal font: regional standard deviation by cruise.
**3.2 Secondary quality control: crossover analysis**
The secondary quality control ($2^{nd}$ QC) method involves comparing the CNR cruises with reference cruises, as
described in Section 2.2. The reference data are considered to be accurate, precise and stable, particularly in deep
water; however, this assumption may not always be effective, especially for recent cruises. There is a notable lack
of comprehensive studies addressing the general trends of dissolved oxygen levels in the Mediterranean Sea.
• *Consistency of the reference data:*
Each profile from the reference cruises (see Table 2) was interpolated using a piecewise cubic Hermite
interpolating scheme to standard pressure values ranging from 0 to 3600 dbar. Subsequently, all profiles were



averaged to produce a single profile for each cruise. Figure 7 illustrates the ratio (A/B) estimates between the tested
cruise (A:6/06MT20011018) and the remaining five reference cruises (B). The results indicate a significant
difference in measurements from the surface to 1000 dbar, which may be attributed to seasonal variations and
atmospheric interactions.
Below 1000 dbar, variability is reduced, with the ratio approaching 1, indicating good agreement among the
cruises, with the exception of cruise 28/11BG20220517 (Fig. 7(e)). This particular cruise exhibited Oxygen levels
approximately ~ 4% lower than those of the other reference cruises, as shown in Fig.4(b).
Observations from this cruise, conducted in 2022, may indicate natural variability in oxygen levels, suggesting a
potential decline in oxygen levels in the WMED, similar to trends observed in other oceanic regions (Grégoire et
al., 2023). However, it is important to consider that this discrepancy could also stem from the precision of the
Winkler titration values and their standardization with potassium iodate (KIO3). A 4% difference in oxygen
concentration is substantial and raises questions about the validity of such variation at these depths.
At depths below 2500 dbar, differences among the reference cruises were noted, revealing similar patterns with
slight variations, that warrant further investigations in subsequent studies. With the exception of cruise
11BG20220517, the deep oxygen measurements exhibited a high degree of alignment across the cruises, indicating
a strong agreement within the depth range of 1000 and 2000 db, where the minimum for transient tracers is
typically located. this layer is characterized by lower temporal variability; thus, any change occuring below 1000
db do not significantly impact the results of the 2$^{nd}$ QC.
In this analysis, we assess the extent to which adjustments should be recommended. Following the standards
established by CARINA and GLODAP data products, no adjustments smaller than 1% for oxygen measurements
were applied (Hoppema et al., 2009). Overall, five out of six reference cruises demonstrated an average ratio below
1000 dbar, within the 1% accuracy limit (ranging from 0.99 to 1.01).
However, the deep average ratio for reference cruise 28/11BG20220517 was recorded at 0.97, indicating that its
oxygen levels were 3% lower. We therefore did not accept this cruise as valid reference data.













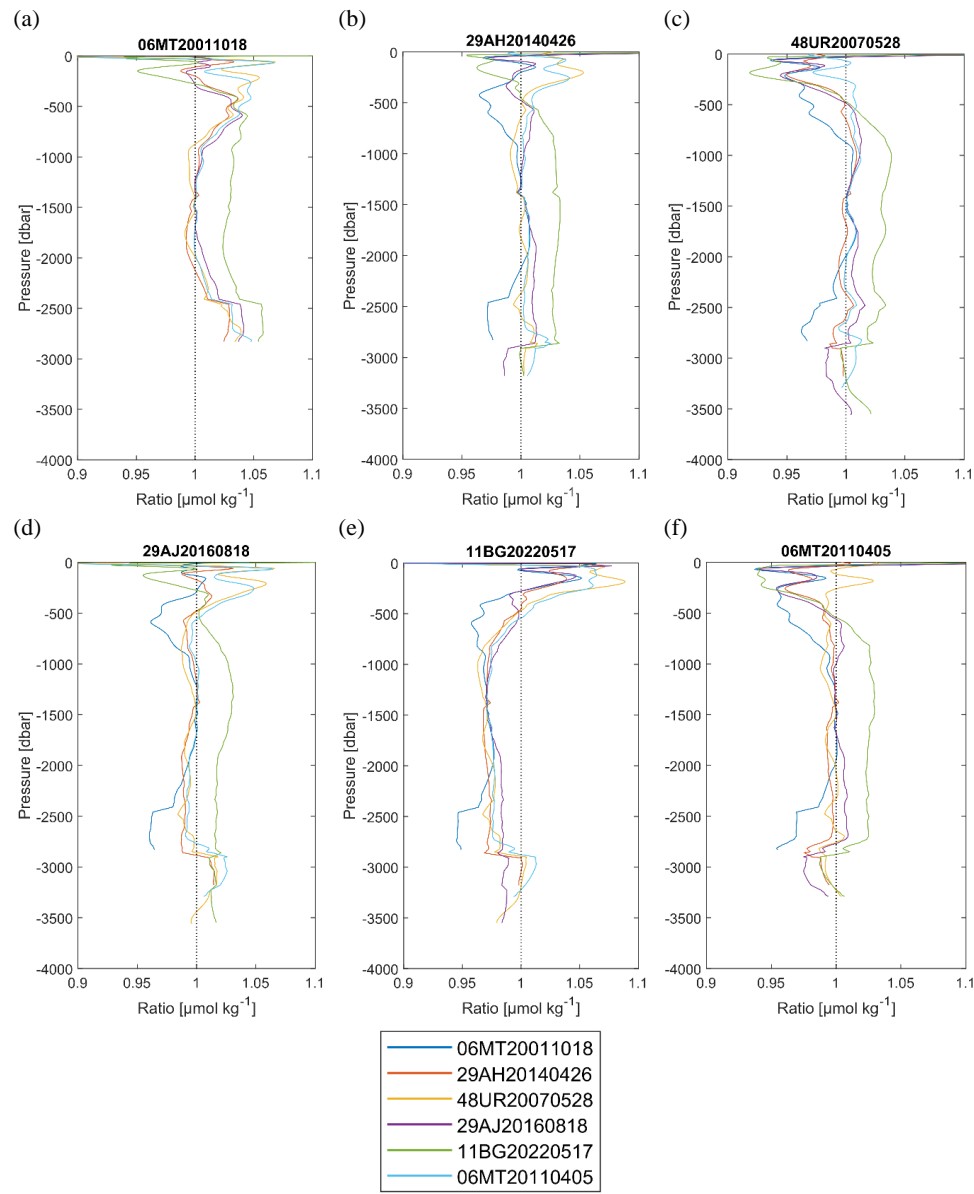


**Figure 7. Vertical distribution of the ratio between the tested reference cruise (top of each subplot) vs the other references (in the legend box below).**




• *Crossover analysis:*
A crossover analysis, following the approach of Johnson et al. (2001) and Lauvset and Tanhua (2015), was
conducted. This analysis is predicated on the comparison of cruise pairs, where the differences between two cruises
within a predefined spatial distance, here a radius of 2° latitude (approximately 222 km) are assessed. In this
process, the interpolated profiles for each station in cruise C1 were compared to the interpolated profiles from
cruise C2 within the specified maximum distance. A difference profile was generated for each pair of stations,
with a minimum of three stations required for each crossover. Calculations were performed on density surfaces to
ensure that data comparisons were made between the same or comparable water masses, thereby mitigating biases
associated with variations in salinity.
Density values were calculated for each measurement, and data from each profile were interpolated using a
piecewise cubic Hermite interpolating scheme to standard density levels. This iterative process was repeated for
each station in the cruise, resulting in multiple difference profiles. The outcome is the weighted mean and standard
deviation of the difference profiles for each cruise, referred to as the offset.  The weighting applied to the profiles
is based on their variability, giving higher importance to parts of the profiles with lower variability (adapted from
Tanhua et al. 2010, 2015). This approach accommodates potential variability in deep layers, particularly in the
Mediterranean Sea, which is influenced by ventilation processes (Testor et al., 2018).
Figure 8 illustrates an example of a crossover and offset between cruise pairs, while an overview of some
crossovers vs. reference dataset are displayed in Section 4. All calculated offsets for each cruise were examined to
determine the presence of any likely biases in the measurements.
Corrections were meticulously reviewed and justified whenever adjustments were deemed necessary (Table 4). It
is important to note that high variability in deep-water of the Mediterranean Sea, particularly in the Northern
WMED where deep convection occurs, may increase the likelihood of detecting offsets in unbiased data.
The number of stations in the overlapping region (i.e., within the predefined radius) is critical for accurate offset
estimation. For instance, as depicted in Figure 8 (a), 27 stations from C1 were compared to 15 stations from C2 to
estimate the offset; a limited number of stations can introduce uncertainty into the offset estimate. Additionally,
while the number of crossover cruises is significant, the Mediterranean Sea has a limited number of reference
cruises available.
Following adjustments, the last step involves evaluation the overall internal consistency of the CNR-O2WMED
using the weighted mean (WM) of the absolute offsets ($D$) of all crossovers ($L$) and the standard deviation ($\sigma$),
following Tanhua et al. (2009) and Belgacem et al. (2020). This assessment quantifies the accuracy of the data
product, as supported by previous studies (Hoppema et al., 2009; Sabine et al., 2010; Tanhua et al., 2009). Notably,
our evaluation is based on offsets relative to a reference dataset, providing a comprehensive understanding of data
consistency.
$\text{WM} = \dfrac{\sum_{i=1}^{L} D(i)/(\sigma(i))^2}{\sum_{i=1}^{L} 1/(\sigma(i))^2}$

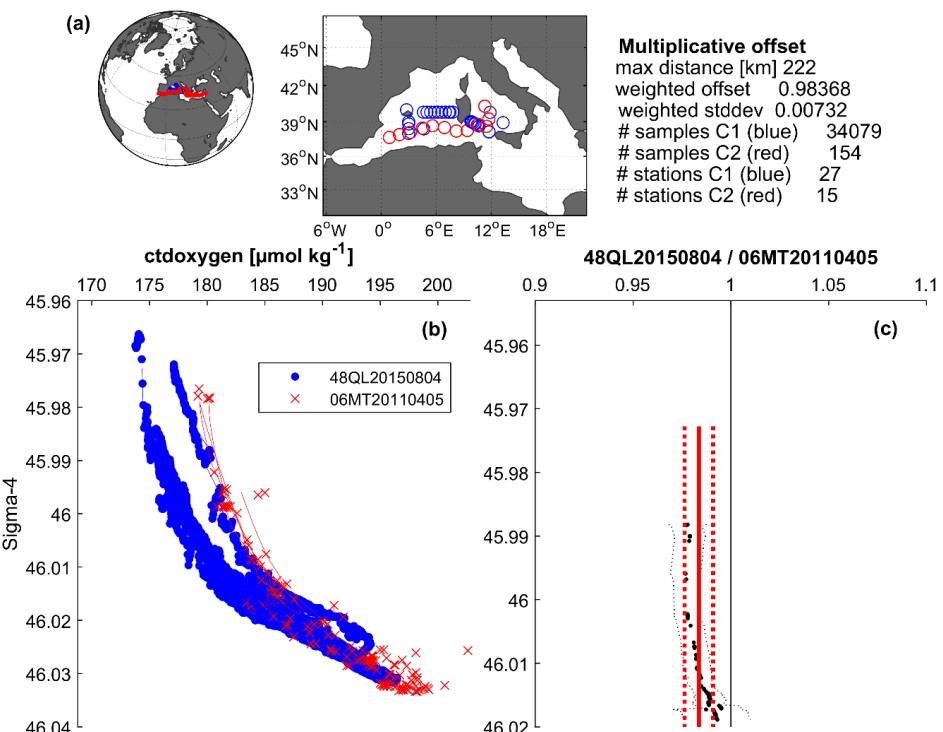

**Figure 8. An illustration showcasing the calculated offset for dissolved oxygen between cruise 48QL20150804 and cruise 06MT20110405 (reference cruise). (a) Spatial distribution of CTD stations involved in the crossover analysis, along with statistical information. (b) Vertical profiles of dissolved oxygen observation (µmol kg⁻¹) from both cruises that fall within the radius of 2° (>1000 dbar). (c) Display of the difference between both cruises (thick dotted black line), standard deviation (thin dotted black lines), the weighted average of the offset (solid red line), and h the weighted standard deviation (dotted red line)**

## 4   Results of the secondary QC and recommendations

The outcomes of the crossover analysis applied to the CNR dissolved oxygen CTD data collected in the WMED are presented in terms of correction factors derived from comparisons with selected reference cruises (GLODAP, CARIMED, MedSHIP). These corrections, aimed at enhancing measurement consistency, are summarized in Table 4. The analysis assumes that the reference dataset represents the true values. This section details the various crossovers, discusses the offsets, and outlines the derived correction factors.

Each crossover was thoroughly evaluated, and corrections were refined as necessary, considering the number of crossovers and the stations involved in each comparison. The offsets and correction factors for each cruise indicated that a significant number of cruises fell outside the predefined accuracy envelope of 1% (as discussed in Section 3.2) and therefore required adjustments. Notably, cruises no. 25, no. 27, and no. 28 were excluded from the crossover analysis due to an insufficient number of stations below 1000 dbar; however, these cruises remain available in the data product.

In total, 73 crossovers were identified. The analysis suggested that deep oxygen measurements from specific cruises (no. 2, no. 8, no. 9, no. 15, no. 17, no. 211, no. 222, and no.23) necessitated upward adjustment when

compared to the reference cruises in the region. Conversely, cruises no. 3, no. 5, no. 21, and no. 24 exhibited
slightly elevated values relative to the respective reference datasets, indicating a need for downward adjustments.
Eight cruises (no. 6, no. 10, no.11/12, no. 13, no. 14, no. 16, no. 18, no. 20, and no.22) did not require any
corrections, demonstrating consistency with the reference dataset and indicating a high quality of the
measurements. A correction is endorsed when the offset exceeds the predefined accuracy envelope of ±1%.
Overall, minor adjustments are proposed, reflecting the overall good quality of the data.
The range of all correction factors was 0.097 (difference between minimum and maximum value), with the most
substantial correction factor reaching 1.05, assigned to cruise 48UR20130604 (no. 211). Correction factors varied
between 0.95 and 0.975 (for adjustments <1) and between 1.018 and 1.05 (for adjustments > 1; refer to Table 4
and Fig. 9).

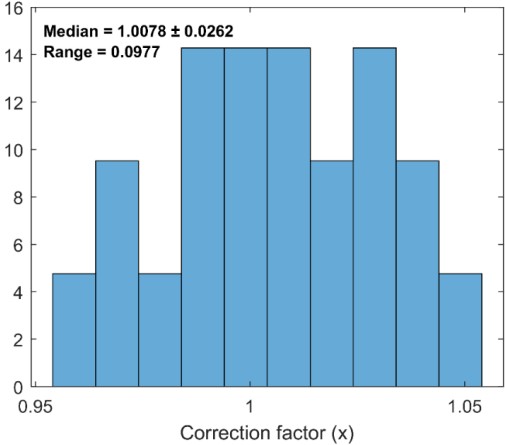


**Figure 9. Distribution of multiplicative correction factor.**
Table 5, Figure 10, and Figure S1 assesses the possible improvements after corrections. In the subsequent text, a
thorough discussion about the adjustments (i.e., Correction factor) of the main challenging cruises is provided.
The reader is invited to compare the descriptions with the respective plots in Figure 10 and the corresponding
crossover summary figures for each cruise.







**Table 4. Summary of the suggested multiplicative adjustments for dissolved oxygen (x) resulting from the crossover**
**analysis.**

| Cruise ID | EXPOCODE | Correction factor ($x$) |
|---|---|---|
| 2 | 48UR20041006 | 1.039 |
| 3 | 48UR20050412 | 0.975 |
| 5 | 48UR20051116 | 0.955 |
| 6 | 48UR20060608 | 1 |
| 8 | 48UR20060928 | 1.031 |
| 9 | 48UR20071005 | 1.025 |
| 10 | 48UR20080318 | 1 |
| 11/12[a] | 48UR20080905/48UR20081103 | 1 |
| 13 | 48UR20090508 | 1 |
| 14 | 48UR20100430 | 1 |
| 15 | 48UR20100731 | 1.034 |
| 16 | 48UR20101123 | 1 |
| 17 | 48UR20110421 | 1.022 |
| 18 | 48UR20111109 | 1 |
| 20 | 48UR20120111 | 1 |
| 21 | 48UR20121108 | 0.973 |
| 211 | 48UR20130604 | 1.052 |
| 22 | 48UR20131015 | 1 |
| 222 | 48QL20151123 | 1.039 |
| 23 | 48QL20150804 | 1.018 |
| 24 | 48QL20171023 | 0.970 |

[a] Cruise #11 and cruise #12 were merged in the secondary QC.
**Cruise no. 2 (48UR20041006)** has crossovers with four references. Notably, the crossover with the reference
06MT20011018 indicated that cruise no.2 had lower values, attributed to the limited number of crossover stations
available for comparison. The remaining three reference cruises demonstrated a consensus regarding a mean offset
of 0.96 (Fig. 10). The offset observed between cruise 48UR20041006 and the reference 06MT20110405 (7 years
difference), 29AH20140426 (10 years difference) and 29AJ20160818 (12 years difference) suggest a 4% increase.
While this increase may appear excessive, it is important to consider that the regional deep averages obtained from
cruise no.2 (Table3) were the lowest. This may provide a rational for the adjustment.

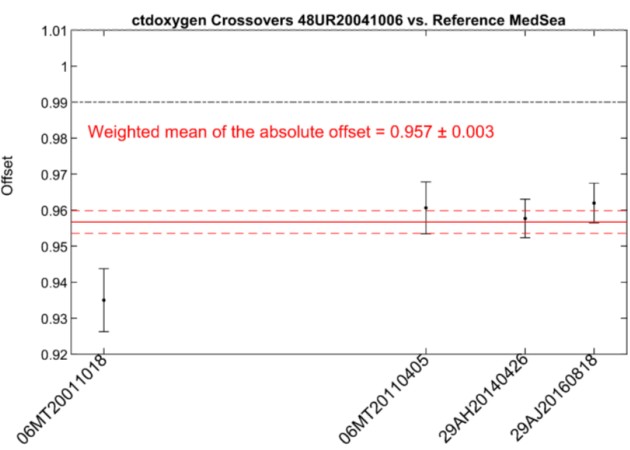


**Figure 10. Summary of offsets for all crossovers found for CTD oxygen on cruise no. 2/48UR20041006. The**
**solid red line indicates the weighted mean of the offsets, with its standard deviation in dashed lines; the**
**dashed grey lines denote the predefined accuracy limits for Oxygen measurements; the black dots with**
**error bars illustrate the weighted mean offsets in relation to individual reference cruises, along with their**
**corresponding weighted standard deviations. The weighted mean and standard deviation of these offsets**
**are annotated within the figure. Note that the reference cruises along the x-axis are arranged in**
**chronological order.**
Data from cruise **no.3 48UR20050412** was compared to the same reference cruises as cruise no.2. The offset
between cruise no.3 and the reference cruises indicates that an adjustment to decrease oxygen is necessary. As
pointed out in section 3.1 cruise no.3 exhibited low precision compared to cruises conducted in the same regions.
Besides, Figures 10 and 11 illustrate similar behaviors with the reference dataset. For this cruise, the offset is
1.025 which supports a downward adjustment of ~3%.

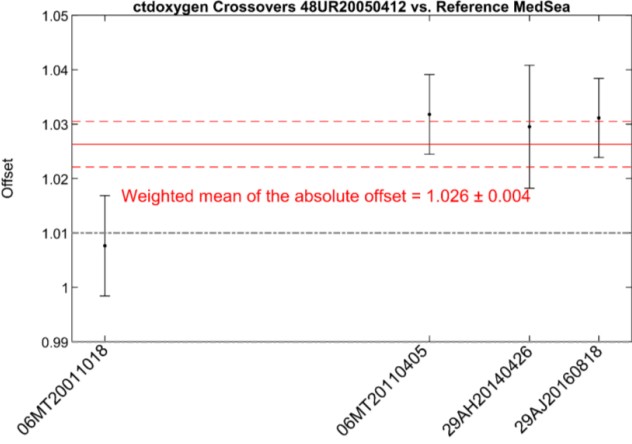


**Figure 11. the same as Fig.10 but for no.3 48UR20050412**
Cruise **no. 5 (48UR20051116)** had two crossovers with the references 06MT20110405 and 29AJ20160818
(Fig.12). An offset of 1.045 is computed. Based on this, an adjustment toward a decrease of ~5% is suggested for
cruise no.5. The discrepancy may indicate potential issues with Winkler values or sensor calibration.

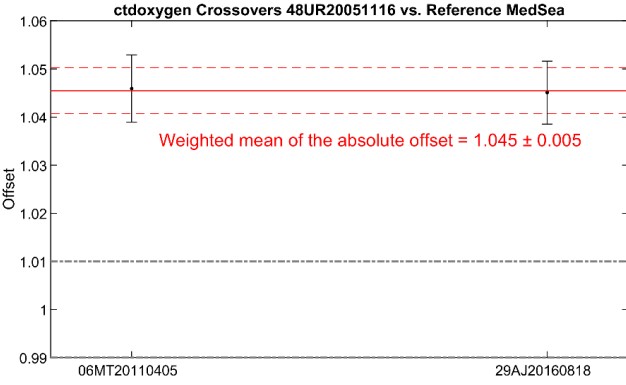


**Figure 12. the same as Fig.10 but for no. 5 (48UR20051116)**
**Cruise no. 6 (48UR20060608)** has five crossovers (Fig. 13). This cruise has few crossover stations with cruise
48UR20070528 which appears to explain the quite large 1.06 offset. The decrease in the measurements is
perceived with crossovers 29AH20140426 and 29AJ20160818 which pointed to similar offset of 1.026 which
means a downward correction of 3%.
While crossovers with 06MT20110405 and 06MT20011018 do not propose any offset, and cruise no.6 seemed to
agree in some parts of the crossing regions;
We refrained from adjusting the data because most of the crossovers did not show consistency and there was good
agreement with the references of years 2001 and 2011.

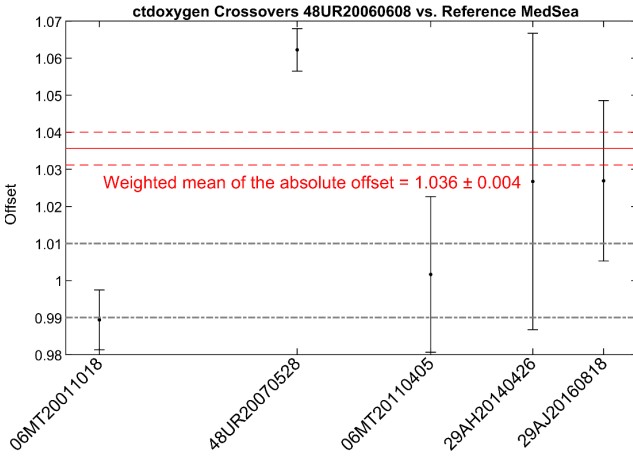


**Figure 13. the same as Fig.10 but for Cruise no. 6 (48UR20060608)**
**Cruise no. 8 (48UR20060928)** has three crossovers in the Tyrrhenian Sea with reference cruises 06MT20110405,
29AH20140426 and 29AJ20160818 (Fig. 14). All three offsets (0.97) point to the same correction toward an
increase of 3%. An adjustment of 1.03 is recommended.

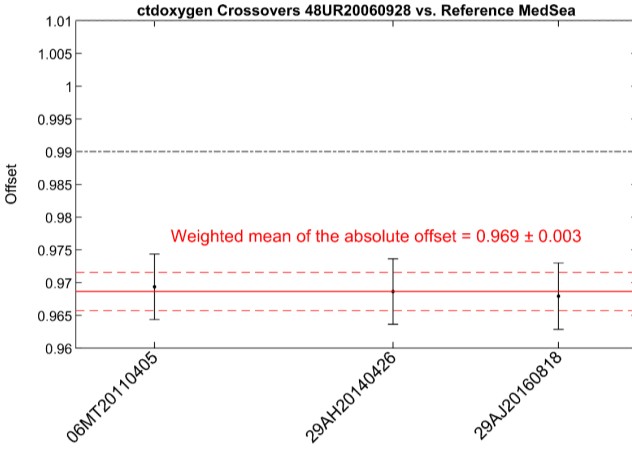




**Figure 14. the same as Fig.10 but for Cruise no. 8 (48UR20060928).**
**Cruise no. 9 (48UR20071005)** has four crossovers. All showing an offset of 0.97 except the crossover with
06MT20011018, where the offset is about 0.95 (Fig. 15). This seems to be large because of the large scatter of the
few stations in the crossing region. Considering the crossovers with 06MT20110405 29AH20140426 and
29AJ20160818, an adjustment of 3% toward an increase is recommended.

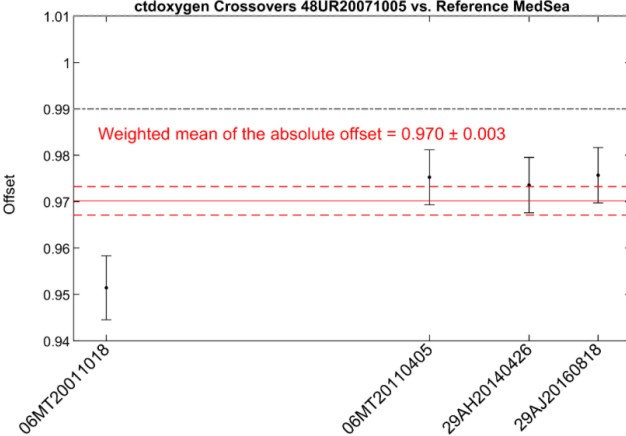


**Figure 15. the same as Fig.10 but for Cruise no. 9 (48UR20071005)**
**Cruise no. 11 (48UR20080905)** has no crossover points with the selected reference dataset. Since it has a two-
month difference from the same year of **cruise no. 12 (48UR20081103)**, both cruises were merged. This 2008
cruise has five crossovers. Crossovers with 06MT20011018 show a large offset of 0.97 which means an increase
of 3%. Though, crossover with 48UR20070528 shows a large offset of 1.03 suggesting a decrease of 3%. Both
crossovers have three crossover points disseminated in different subregions. Their offsets are noticeably higher
when compared to the offsets observed in crossovers with the reference cruises 06MT20110405, 29AH20140426
and 29AJ20160818 (Fig.16). However, cruise no.11/12 conducted in 2008, consistently shows no significant
offset. Based on this evidence and the inconsistency in the 2001 and 2007 crossovers, we decided for no
adjustments

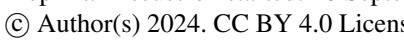


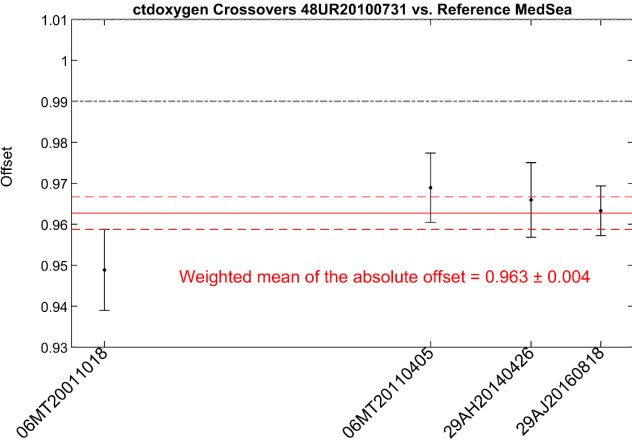


**Figure 16. the same as Fig.10 but for Cruise no. 11 and no. 12.**

**Cruise no. 15 (48UR20100731)** has four crossovers. Crossover with 06MT20011018 show very large offset of
0.95 suggesting 5% increase. With few stations in the crossing regions, this crossover is not warrant of reliable
adjustment.
Crossovers with 06MT20110405 (one year difference), complies with the singular crossovers with the reference
29AH20140426 (4 years difference) and 29AJ20160818 (six years difference) about an offset of ∼0.97 (Fig.17),
which gives further evidence for an adjustment of 3% increase.
**Figure 17. The same as Fig.10 but for Cruise no. 15 (48UR20100731).**
**Cruise no. 17 (48UR20110421)** has two crossovers in the Tyrrhenian Sea both agree about an offset of 0.97
(Fig.18). Based on this, an adjustment of 2% toward an increase is suggested. Crossover with same year reference
cruise 06MT20110405 gave offset of 0.97. Same offset and adjustments are suggested to cruise no. 8
(48UR20060928) in the same region. This demonstrates additional evidence about the suggested adjustment of
cruise no.17, to make it consistent with the neighboring cruises.

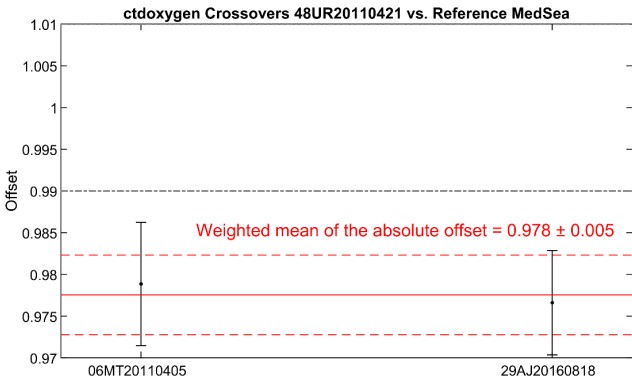

**Figure 18. The same as Fig.10 but for Cruise no. 17 (48UR20110421)**
**Cruise no. 18 (48UR20111109)** has four crossovers. Offsets with the same year reference cruises 06MT20110405
are in good agreement, same with offsets computed with 29AH20140426 (3 years difference) and 29AJ20160818
(5 years difference) that were 0.99 (Fig. 19). Except with crossover with 06MT20011018 (10 years difference)
suggestion an offset of 0.97 and an adjustment of 2% increase. Few crossover stations are used to estimate the
offset which explains the large offset compared to the other reference. Hence, we have concluded to refrain from
making any adjustments.

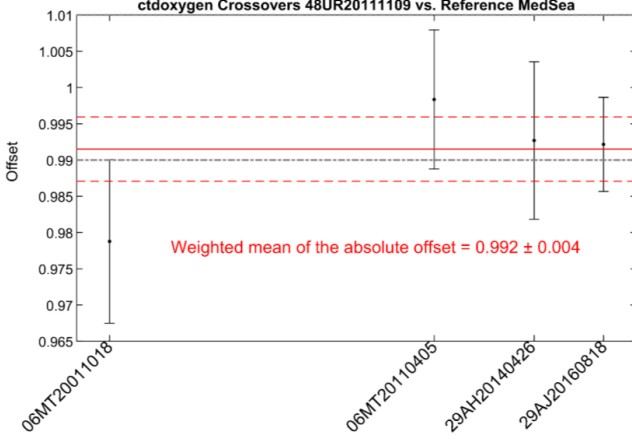

**Figure 19. the same as Fig.10 but for Cruise no. 18 (48UR20111109).**
**Cruise no. 21 (48UR20121108)** has three crossovers with the reference cruises (Fig. 20). In view of crossovers
with 06MT20110405 (one year difference), an offset of 1.026 is found suggesting that the data appear high than
the reference and require a downward adjustment of ~3%. The adjustment appears justified because the offset
with the references 29AH20140426 and 29AJ20160818 suggest similar offset magnitude. Consequently, the
adjustment was set to a decrease of 3%.

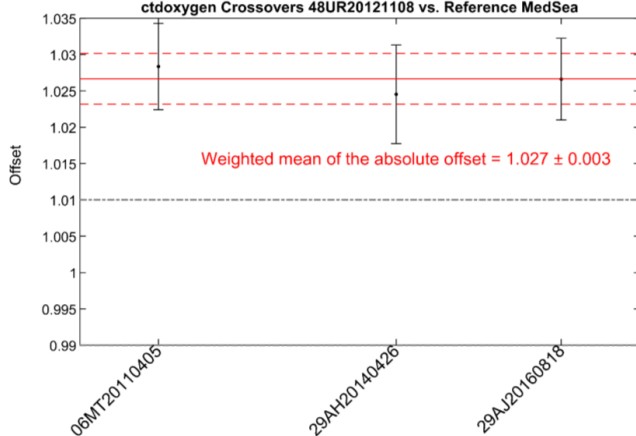


**Figure 20. The same as Fig.10 but for Cruise no. 21 (48UR20121108).**
**Cruise no. 211 (48UR20130604)** has an offset of 5% of similar magnitude in all crossovers with the four reference
cruises 06MT20011018, 06MT20110405, 29AH20140426 and 29AJ20160818 (Fig. 21). Cruise **48UR20130604**
seems to be high. Connecting it to cruise **no. 22 (48UR20131015)** from the same year, this cruise is in good accord
with the same reference cruises in the same crossing areas and did not show any offset, providing additional
evidence. We think that an adjustment of 5% toward an increase is justified to bring the data within the acceptable
range.

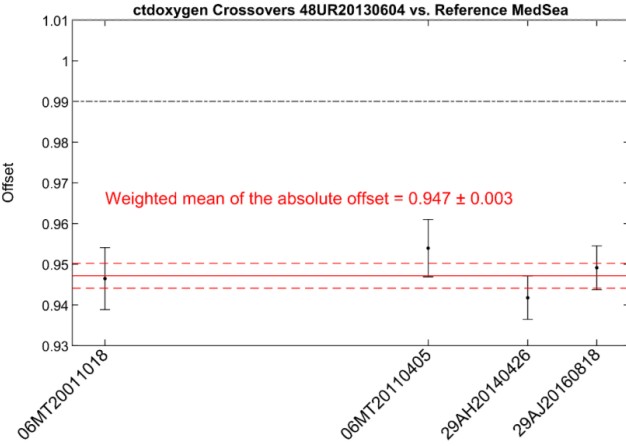


**Figure 21. The same as Fig.10 but for Cruise no. 211 (48UR20130604).**
**Cruise no. 222 (48QL20151123)** has a consistent offset of 0.96 with all three crossovers spanning different years
with cruises 06MT20110405 (four-year difference), 29AH20140426 (one year difference) and 29AJ20160818 (1



year difference) (Fig. 22). Deep measurement from this cruise seems to be lower than the reference suggesting a
correction of 4% toward an increase.

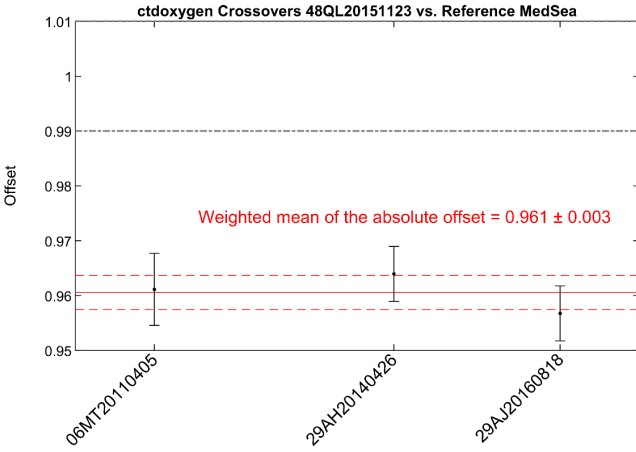


**Figure 22. The same as Fig.10 but for Cruise no. 222 (48QL20151123).**

Similar increase is apparent in **Cruise no. 23 (48QL20150804)** happened in the same year 2015. Three crossovers
with the references 06MT20110405, 29AH20140426 and 29AJ20160818 agree about a ∼2% increase based on a
mean offset of 0.98 (Fig. 23), whereas crossover with the reference cruise 06MT20011018 (14 years difference)
propose an offset of 0.96, accordingly 4% increase, which is very high and not enough justified. We therefore
suggest a correction following the three references.

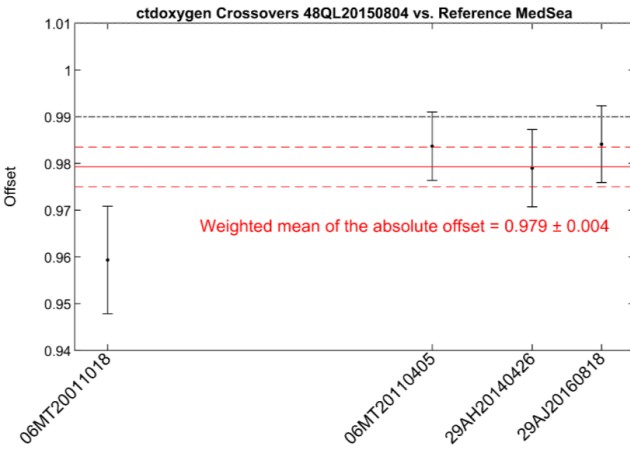


**Figure 23. The same as Fig.10 but for Cruise no. 23 (48QL20150804).**

And finally, **cruise no. 24 (48QL20171023)** has three crossovers (Fig. 24) with the reference 06MT20110405 (six
years difference), 29AH20140426 (three years) and 29AJ20160818 (one year difference). A constant offset of
1.029 is assessed indicating that cruise no.24 is higher than the reference. The offset appears to be persistent in all

these years and is suggestive of an adjustment of 3% toward a decrease, that should be appropriate. We decided to

follow the suggestion.

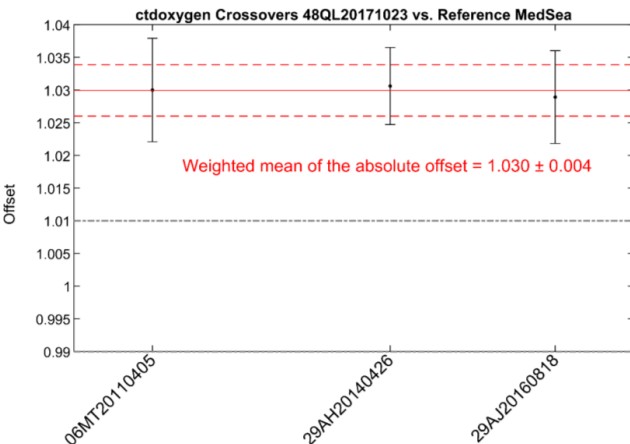

**Figure 24. The same as Fig.10 but for Cruise no. 24 (48QL20171023).**

To validate our findings, we recalculated the offsets using the adjusted data after applying the corrections outlined

in Table 4.

Following the application of these adjustments, the offsets were reduced. The suggested cruises now fall within

the accepted envelope of 1% indicating an enhanced consistency of the measurements. This improvement is

evident in the adjusted data presented in Figure 25 (in blue) and table 4.

To evaluate the consistency, we computed WM of the absolute offsets. The consistency of the adjusted oxygen

dataset experienced a slight enhancement, increasing from 0.991 to 0. 998%. This corresponds to an improvement

in the internal consistency of the dataset by 0.7%. The minor improvement in consistency, coupled with the

reduced range, underscores the high quality of the initial dataset and efficacy of the quality assurance procedures

implemented during each cruise.

The adjustments removed potential biases arising from errors related to measurement, calibration, data handling

practices, and the lack of adherence to international standards improving the overall consistency.

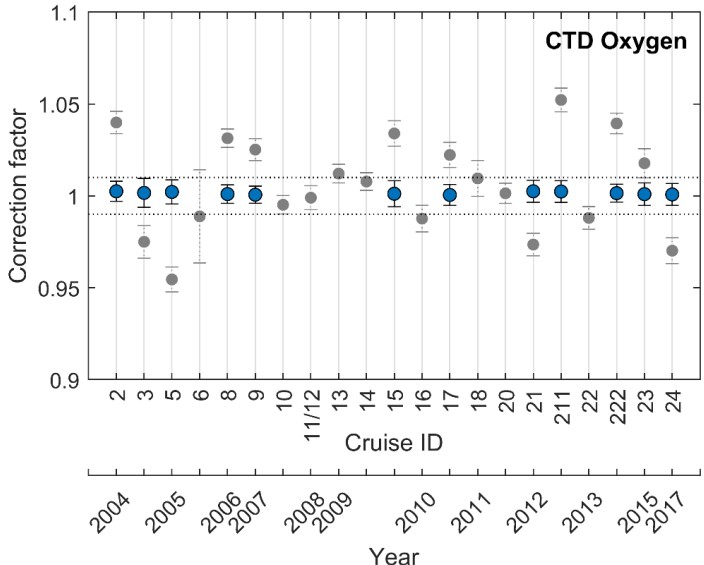

479

**Figure 25. Results of the crossover analysis results for CTD dissolved oxygen, showing the correction before (in grey) and after (in blue) adjustment. Error bars indicate the standard deviation of the absolute weighted offset. A correction means the original CTD Oxygen data must be multiplied by that amount (see Table 4). The dashed line represents the 1% accuracy envelope for an adjustment to be made.**

**Table 5. Results from the Secondary QC: improvements of the weighted mean of absolute offset per cruise for both unadjusted and adjusted data. "n" represents the number of crossovers per cruise. Values in bold (>1) signify instances where the measurements from the tested cruises are lower than the reference data.**

| Cruise ID | EXPOCODE | CTD Oxygen (%) | | | Crossover regions |
|---|---|---|---|---|---|
| | | $n$ | Unadjusted | Adjusted | |
| 2 | 48UR20041006 | 4 | **0.96** ±0.006 | 0.99±0.005 | Tyrrhenian sea<br>Algerian basin<br>Alboran sea |
| 3 | 48UR20050412 | 4 | 1.025±0.008 | 0.99±0.007 | Tyrrhenian sea<br>Sicily-Sardinia Channel<br>Algerian basin<br>Alboran sea |
| 5 | 48UR20051116 | 2 | 1.045±0.006 | 0.99±0.006 | Tyrrhenaian Sea<br>Sicily-Sardinia Channel |
| 6 | 48UR20060608 | 5 | 1.01±0.02 | NA | Ligurian Sea<br>Tyrrhenain Sea<br>Algerian basin<br>Saridinia-Balearic sea |
| 8 | 48UR20060928 | 3 | **0.96**±0.005 | 0.99±0.005 | Ligurian Sea<br>Tyrrhenain Sea<br>Algerian basin<br>Saridinia-Balearic sea |
| 9 | 48UR20071005 | 4 | **0.97**±0.005 | 0.99±0.004 | Tyrrhenain Sea<br>Algerian basin<br>Saridinia-Balearic sea |
| 10 | 48UR20080318 | 2 | 1.004±0.005 | NA | Tyrrhenain Sea |
| 11/ 12 | 48UR20080905/ 48UR20081103 | 5 | 1.001±0.006 | NA | Tyrrhenaian Sea<br>Sicily-Sardinia Channel<br>Algerian Basin<br>Alboran Sea |
| 13 | 48UR20090508 | **3** | **0.989**±0.005 | NA | Tyrrhenian Sea |





| | | | | | Sicily-Sardinia channel |
|---|---|---|---|---|---|
| 14 | 48UR20100430 | **4** | **0.99**±0.004 | NA | Tyrrhenain Sea<br>Algerian basin<br>Saridinia-Balearic sea<br>Sicily-Sardinia channel |
| 15 | 48UR20100731 | 4 | **0.96**±0.006 | 0.999±0.007 | Tyrrhenian Sea<br>Sicily-Sardinia channel<br>Algerian basin<br>Alboran Sea |
| 16 | 48UR20101123 | 2 | 1.012±0.07 | NA | Tyrrhenian Sea |
| 17 | 48UR20110421 | 2 | **0.97**±0.006 | 0.999±0.005 | Tyrrhenian Sea<br>Sicily-Sardinia channel |
| 18 | 48UR20111109 | 4 | **0.99**±0.009 | NA | Tyrrhenain Sea<br>Algerian basin<br>Saridinia-Balearic sea |
| 20 | 48UR20120111 | 4 | **0.99±±0.005** | NA | Tyrrhenain Sea<br>Sicily-Sardinia channel<br>Algerian basin<br>Saridinia-Balearic sea |
| 21 | 48UR20121108 | 3 | 1.026±±0.006 | 0.99±0.006 | Tyrrhenain Sea<br>Saridinia-Balearic sea<br>Sardinia channel |
| 211 | 48UR20130604 | 4 | **0.950±±0.006** | 0.997±0.005 | Tyrrhenain Sea<br>Sicily-Sardinia channel<br>Algerian basin<br>Saridinia-Balearic sea |
| 22 | 48UR20131015 | 4 | 1.012±0.006 | NA | Tyrrhenain Sea<br>Sicily-Sardinia channel<br>Algerian basin<br>Saridinia-Balearic sea |
| 222 | 48QL20151123 | 3 | **0.96**±0.005 | 0.998±0.004 | Tyrrhanaian Sea<br>Sicily-Sardinia channel |
| 23 | 48QL20150804 | 4 | **0.98**±0.007 | 0.999±0.006 | Ligurian Sea<br>Tyrrhenain Sea<br>Algerian basin<br>Saridinia-Balearic sea |
| 24 | 48QL20171023 | 3 | 1.029±±0.007 | 0.99±0.005 | Tyrrhanaian Sea<br>Sicily-Sardinia channel |


## 5 Summary and conclusions

This study's main objective was to enhance data quality, assurance, and accessibility within the context of FAIR
data principles. Acknowledging the limitations of the reference datasets is essential; although considered high
quality, they are not infallible and may not fully cover the entire temporal range of interest. Regardless, our quality
control efforts have yielded positive results, as demonstrated throughout this study.
The majority of the CTD oxygen data now falls within the predefined acceptance range of 1%, indicating a
consistent and accurate dataset. This aligns with established standards seen in widely used datasets, such as
CARINA and GLODAP (Hoppema et al., 2009, Tanhua et al., 2009). Furthermore, the adjustments made to
address systematic biases between reference datasets and the CNR cruises have significantly improved the internal
consistency of Oxygen measurements.
Despite the inherent challenges in assessing changes in oxygen levels due to the scarcity of measurements, the
recent oxygen data contribute substantially to our understanding of oxygen variability in the region.
The CNR-O2WMED dataset serves as a valuable regional resource, providing quality-controlled measurements
that facilitate accurate trend quantification and estimation of changes, thereby making it an essential tool for future
studies with acceptable temporal and spatial coverage, and the potential extension of the analysis to the Eastern
Mediterranean Sea. The utility of this dataset extends to its assimilation into ocean models and the verification of
regional models, offering critical insights in the oxygen cycle.
As we face warming ocean and increased stratification, the anticipated decline in oxygen levels may lead to
enhanced acidification and reduced $CaO_3$, potentially accelerating in the remineralization of organic matters at
shallower depths. Consequently, the expansion of low oxygen zones is expected, posing significant ecological
challenges (Keeling and al., 2009).
While the current dataset does not incorporate data from various sources to avoid potential inconsistencies,
ensuring the reliability of the CNR-O2WMED data remains a priority before any future integrations.  The dataset's
utility will be maximized when used in conjunction with other data sources, such as BGC-Argo and glider data,
allowing for more comprehensive analysis of the oxygen dynamics in the WMED.
Future studies should focus on synergizing regional datasets to enhance our understanding of ongoing changes in
Biogeochemical cycles. This collaborative approach will ultimately contribute to global efforts to monitor and
mitigate the impacts of ocean deoxygenation, fostering a deeper understanding of biogeochemical processes and
their implications for marine ecosystems.
Future studies should focus on synergizing regional datasets to enhance our understanding of the ongoing changes,
ultimately contribution to global efforts to monitor and mitigate the impact of ocean deoxygenation.

## 6    Data availability

The CNR_O2WMEDv1 (Belgacem et al., 2024 [in review], see temporary link below) dataset is available at
PANGAEA (submitted on 21/08/2024, DOI in preparation). It consists of two parts; the first is the original data
product which has undergone both calibration and 1st quality check. The second is the original product adjusted
using the recommended corrections from the secondary quality control. The dataset is complementary to the data
product CNR-DIN-WMED available https://doi.org/10.1594/PANGAEA.904172. No special software is required
to access the data.
**Temporary link to CNR_O2WMED_ODV format:**
**https://cnrsc-**
**my.sharepoint.com/:f:/g/personal/malekbelgacem_cnr_it/EkIgo958UMlBmJ8SGwNB4HwBBX-**
**FzDa8Nl9C3vHeS4Vd4Q?e=Wt8p1I**
Table 6 summarizes the list of parameters included.
**Table 6. Summary of data product parameters and units.**

| Variable | Data Product file parameter name | Data product WOCE flag name | Units |
|---|---|---|---|
| Expedition/cruise code | EXPOCODE | | |
| Cruise ID | CRUISE | | |
| Station number | STNNBR | | |
| Year | YEAR | | |
| Month | MONTH | | |
| Day | DAY | | |
| Latitude | LATITUDE | | decimal degree |



| Longitude | LONGITUDE | | decimal degree |
|-----------|-----------|---|----------------|
| Pressure | CTDPRS | | decibar |
| Temperature | CTDTMP | | °C |
| Salinity | CTDSAL | CTDSAL_FLAG_W | |
| Oxygen | CTDOXY | CTDOXY_FLAG_W | $\mu mol\ kg^{-1}$ |



## Authors contributions

MaB ran the analysis and wrote the manuscript. KS contributed to writing the manuscript. MA, SKL contributed
to the analysis. JC contributed to specific parts of the manuscript. MiB and SS coordinated the technical aspects
of most of the cruises. CC and TC assisted some of the chemical analysis and contributed to write the analytical
method.

## Competing interest

The authors declare that they have no conflict of interest.

## Acknowledgements

The data have been collected in the framework of several national and European projects, e.g. KM3NeT, EU GA
no. 011937; SESAME, EU GA no. GOCE-036949; PERSEUS, EU GA no. 287600; OCEAN-CERTAIN, EU GA
no. 603773; COMMON SENSE, EU GA no. 228344; EUROFLEETS, EU GA no. 228344; EUROFLEETS2, EU
GA no. 312762; JERICO, EU GA no. 262584; and the Italian PRIN 2007 program "Tyrrhenian Seamounts
ecosystems" and the Italian RITMARE flagship project, both funded by the Italian Ministry of Education,
University and Research.  The authors thank the Horizon Europe – IA REDRESS project for the funding.
The authors are deeply indebted to all investigators and analysts who contributed to data collection at sea during
so many years as well as to the PIs of the cruises (Stefano Cozzi, Gabriella Cerrati, Stefano Aliani, Mario Astraldi,
Maurizo Azzaro, Alberto Ribotti, Massimiliano Dibitetto, Gian Pietro Gasparini, Annalisa Griffa, Jeff Haun, Loïc
Jullion, Gina La Spada, Elena Mannini, Angelo Perilli and Chiara Santinelli), the captains, and the crews for
allowing the collection of this enormous dataset; without them, this work would not have been possible.

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
