# Peer review of "A consistent regional dataset of dissolved oxygen in the Western Mediterranean Sea (2004-2023): CTD-O2WMED"

_Earth System Science Data, 2024_

## Author Comment (AC1)

**Response to comments from RC2**

On behalf of all authors, we would like to thank the reviewer for their reading of the manuscript and their remarks. Your comments provided valuable insights to refine and clarify the manuscript. We have taken into consideration all suggestions.

In the following, we try to address all issues raised as best as possible.

Below, the reviewers' comments are given in *italic blue* and responses in normal black font.

Belgacem and coll. present a compilation of dissolved oxygen profiles obtained in the western Mediterranean. This dataset covers the period 2004 - 2023, although most of the data were collected between 2004 and 2013. The introduction is brief, as are the methods and results sections, while the article presents the data qualification in great detail. It's true that a data paper should explain the data and quality controls, but in the end this version of the submitted article mainly covers this aspect. What is missing, however, is a contextualization of the topic (oxygenation of Mediterranean waters) based on recent articles, a more detailed description of the data set, and a comparison of the data with other data sets (Argo, other campaigns in the Mediterranean).

• We agree that placing our dataset in a broader scientific context is important. Accordingly, we have significantly expanded the introduction to include a more detailed discussion of oxygen dynamics in the Mediterranean Sea, highlighting recent findings on deep convection and oxygen variability. We incorporated the suggested references (Fourrier et al., 2022; Ulses et al., 2021).

Many details are given for quality control. But there are almost no details about how the data was collected. It is only written that (line 39) 1382 CTD oxygen profiles, but there is no information about how they were obtained. This is a real problem because to use a dataset, it is important to know the type of sensors used, their accuracy, how they are calibrated and the calibration frequency, especially for marine waters where the values must be very precise.

• We have added mor details about calibration. This information is now included in the Methods section to provide users with a clear understanding of data provenance and reliability.

It would also be useful to explain why there have been far fewer measures since 2014, and to indicate whether this will continue.

• Th reduction in data collection after 2014, notably due to the loss of the main research vessel *URANIA* in 2015, which affected the continuity of measurements. We comment on the prospects for future data collection efforts.

**It is not explained also what are reference cruises (figure caption 2) and how they were defined.**

• We have clarified the definition of reference cruises in the revised version and main text, explaining their role as high-quality baseline datasets used for sensor calibration and quality control.

The introduction to Dissolved Oxygen (DO) and the Mediterranean Sea is minimalist, with a few generalisations about this element and some very conventional references, and then very few details about DO dynamics in this sea, which is considered a hot spot for climate change. To justify the interest of the database presented, a more precise state of the art is

required. Some work has already been published, in particular on the DO minimum, which is strongly determined by the intermittent convection process. There is an important issue related to the decrease in the intensity of deep convection in the northwestern Mediterranean predicted by the end of the century in recent projections, which may have important consequences on the DO concentration. This needs to be highlighted in the article, as it reinforces the interest in making DO datasets available in the Mediterranean Sea. The Mediterranean Sea is not my usual area of study. However, I would like to suggest some recent articles :

Fourrier & al (2022) Impact of Intermittent Convection in the Northwestern Mediterranean Sea on Oxygen Content, Nutrients, and the Carbonate System JOURNAL OF GEOPHYSICAL RESEARCH-OCEANS DOI10.1029/2022JC018615

Ulses e& al (2021) Oxygen budget of the north-western Mediterranean deep-convection region BIOGEOSCIENCES DOI10.5194/bg-18-937-2021

• We have expanded and updated the state-of-the-art discussion in the introduction, integrating recent literature on dissolved oxygen dynamics, including the importance of intermittent convection, projected changes under climate scenarios, and implications for Mediterranean oxygen budgets. This reinforces the motivation for making quality-assured oxygen datasets publicly available.

The description of the quality assurance method needs to be improved, the principle of the method is relevant to be described in the article with an example, but the other cases should be placed as a supplementary section or provided as information at the link where the data can be accessed.

• We have improved the description of the control methods, including a detailed explanation of the approach applied to the new database. While the core principles and representative examples are now in the main text, for more detailed cruise-specific quality control cases ,plots are accessible upon request for now.

**Even though it is a data paper, the data need to be better presented, how they compare with other data sets or Argo profiles.**

• We are comparing with other data sets: the reference cruises are different. But we will take that into consideration: a dedicated section about comparison of the new product with other cruises is implements at selected sub regions. Besides the accuracy of the Argo float data is generally lower than that of CTD profile, especially with the recent issues encountered with BGC-Argo. The dataset will be used together with other data sources and shall be compared with other globally-averaged O2 climatology, which is not the scope of this paper. But we will do in future work.

The conclusion also needs to be considered, for example, it is underlined (lines 498) that the accuracy of the measurements, but in fact the article does not introduce what data are available for this region.

• The purpose of the paper is to make available the Italian cruise data. We added more information about the already existing data.

The text in the article also needs to be improved. For example, it is strange to start an abstract by mentioning the doi of another dataset (lines 16-18 lines). This sentence can appear in the body of the article, but not in the abstract.

• We revised the abstract to remove the DOI and moved this information to the data availability section, improving readability and flow.

In summary, I have very mixed feelings about this paper submitted by Belgacem and coll. On the one hand, it presents an interesting dataset, but at this stage it needs a thorough rewrite.

• We believe that the revised manuscript now addresses your concerns and better meets the expectations for a comprehensive and well-contextualized data paper. We thank you again for your valuable comments and hope the improved manuscript will be of interest to the scientific community.

---

## Author Comment (AC2)

**Response to comments from RC1**

On behalf of all authors, we would like to thank the reviewer Dr. Toste Tanhua for their thorough reading of the manuscript and their constructive remarks and suggestions. Your comments provided valuable insights to refine and clarify the manuscript. We have taken into consideration all suggestions.

In the following, we try to address some issues raised, all comments and suggestions were implemented in the new version as best as possible. For a full overview , please refer to the undated manuscript version.

Below, the reviewers' comments are given in *italic blue* and responses in normal black font.

*The manuscript from Belgacem et al. is setting out to produce a data product focusing on consistent ocean oxygen values in the western Mediterranean Sea, with some important restrictions, such as only considering data collected on Italian vessels, and only considering the oxygen data from the CTD sonde. It is a well needed attempt to move toward a consistent set of ocean data, fit to determine variability and trends in oxygen in this particular area.*

- We appreciate your recognition of the value of our effort to establish a consistent oxygen dataset for the WMED region. We have clarified in the manuscript the rationale behind restricting the dataset and focusing on CTD sensor data, emphasizing data quality and methodological consistency

*The abstract promises "The quality assurance process involves calibration of CTD measurements against Winkler analyses and the comparison of deep observations with reference datasets, using the crossover analysis". However, I see only weak connections in the manuscript between Winkler and sonde (CTD) data. The supplementary information is very short, only 2 tables. These are of interest and I suggest including this information in the main manuscript rather than having a SI. Maybe the information from table S1 could be included in tables 1, for instance. There are a lot of details on how the analysis of Winkler and CTD data is done that needs to be discussed in the manuscript. Table S1 has no information on any post-calibration of the CTD sensor: Was a "post-calibration" needed? If so, how was this done, and how large was the post-calibration?There needs to be a discussion on the difference of this analysis, that uses CTD sensor data, vs. the Winkler data.*

- We have added a dedicated section in the revised manuscript describing the post-calibration assessment. Moreover, we expanded the discussion on the differences between CTD sensor data and Winkler titrations, highlighting the strengths and limitations of each and their complementary roles in our quality control process.

*It took me a while to understand that the manuscript is about the CTDO data, and not about the discrete bottle data, as is the case for CARIMED and GLODAP. Also, and importantly, if there was an adjustment applied to the CTDO data, was the same adjustment applied to the Winkler data for that cruise?*

- We have clarified early in the manuscript that our focus is on CTD oxygen sensor (CTDO2) profiles rather than discrete bottle data. Because Winkler titrations were limited in number, we did not apply the same adjustment to Winkler data; instead, Winkler measurements served as independent calibration points for the CTD sensors. This distinction and rationale are now explicitly stated.

*And where is this documented, and where are the bottle data for those cruises? Actually, where are the bottle data for all of these cruises? This connection is particularly important for any*

*future effort to make a data product of oxygen data in the Mediterranean Sea that is more universal, but based on bottle data.*

- Metadata including information about bottle data for each cruise is now provided in Cruise report link the Supplementary Material, the data is not yet available. We acknowledge the importance of integrating bottle data for a more universal Mediterranean oxygen product and have noted that gathering accessible Winkler datasets is a key future goal.

*The link to the temporary data set did not work, so I could not inspect the product. This in itself is a disqualifier for publication, that can only happen once the data is on a recognized repository.*

- We apologize for the inconvenience. The link is now active and points to the updated dataset currently under review at PANGAEA: https://doi.pangaea.de/10.1594/PANGAEA.974725. The dataset will be publicly available soon.

*Minor issues:*

*Abstract: Quality assurance is the process of checking the data while doing the measurements, so that you can react and correct any issues you notice. What you have done here is quality control, not quality assurance.*

- Done: We revised the terminology to use "quality control" consistently

*Line 38. Is that the end of the sentence? Add period.*

*Line 39: Oxygen is not leading to high productivity, but is the result of high (primary) productivity.*

- Done: Corrected the sentence accordingly.

*Line 41: OMZs are certainly important, but the OMZ in the Med is very weak, i.e. most areas of the (open) Med is well oxygenated. I suggest to note this in the text. OMZs cannot become "more frequent", as they the term OMZ refer to an area, not an event. OMZ can become larger/smaller, more, or less, intense, but not more/less frequent. Low oxygen events can, however, become more frequent.*

- Done: We clarified the limited extent and intensity of Mediterranean OMZs and rephrased to reflect that low oxygen events, rather than OMZ frequency, may increase.

*Table 1: Maybe add a column on which oxygen sensor was used, and if there was a post-calibration needed after comparison with Winkler data.*

- We included information about  post-calibration.

*Lines 132-135: As Far as I know, the CARIMED data product is not published, or final. How can you be sure that the oxygen has "reference cruise" quality by simply stating that they were part of a non-ready product? Also, did you make a check on consistency between Winkler data and CTD data for the reference cruises as well, noting that CARIMED is only bottle data`*

- Done: We clarified the reference cruises were evaluated using Langdon et al. (2010) methodology with certified reference materials (OSIL KIO3) and note the CARIMED product is preliminary.

*Would there be another, more objective, way of assigning reference cruises, such as using cruises that used CRMs or certified standards?*

- To our knowledge he references cruises used the Langdon (2010) methodprecisely they used OSIL 0.01N KIO3 to standardize the thiosulfate; this is a reference material about CARIMED

*Lines 181-219, roughly. Most paragraphs is this section is only one sentence long. It makes it awkward to read. Consider modifying. Much of this information is probably better in a table.*

*Line 227: Why is this an issue for recent cruises?*

- Added explanation

*Line 232: It seems very strange to average each cruise to one single profile. I cannot see how that can work, except for cruises with limited geographic extent. Why not make an average profile for each of your sub-regions instead, that might work out.*

- Done: We now provide regional mean profiles in addition to cruise averages to better represent spatial variability.

*Line 269: Section crossovers. I did not understand if you made a cross-over analysis of all cruises vs. each other, or only the cruises in the product vs. the reference cruises. The text is not clear on that.*

- Done: The text now clearly states that crossover analysis was performed between all cruises and reference cruises, with a detailed explanation of the methodology.

*Section 4: I would strongly suggest to use coherent definitions, and standardized language for the description of the adjustments to each cruise. I would suggest to make sure the ms is consistent in using "offset" – this is the difference between a cruise and reference cruises: "correction" – this is the reciprocal of the offset and indicate what would need to be done to get consistency based on the determined off-sets – "Adjustment"- this is what you did to the cruise based on different lines of evidence.b Reading the text on each cruise, it is not always clear what you did to the cruise due to fluffy language. I would recommend not to to fancy on language, but be consistent and clear.*

- Done: Terminology was harmonized for clarity and consistency.

*There are many many figures to the section describing the individual cruises. Maybe having one or two cruises with many panels would be much easier to read .*

- Done: We reduced the number of panels per figure and focused on two representative cruises for detailed display.

---

## Author Response (AR2)

TOSTE MINOR REVISION : To be checked by Editor

We Thank the reviewer and editor for the comments and suggestions.

We found some issue locating the exact line number that the reviewer is referring to. We tried to make the correspondence. We hope all suggestions been tackled (response in red).

I thank the authors for addressing issues in the first round of revisions.
However, there are still some issues that needs to be tended to, as detailed below.

In addition, there are numerous small language errors and editorial glitches, such as incomplete sentences. I recommend one more round of careful language check.

I thin this manuscript needs a minor revision. Most important is the inconsistencies in the data products available at PANGAEA with what is stated in the manuscript, as the unclear units for salinity and state of adjustements or not in the data product.

Detailed listing:
Line 170: Complete sentence
Line 180: Please explain why you do not attempt to "post calibrate" the CTD-oxygen values from winkler for those cruises where you see a large offset.

Line 224: How can vertical diffusion from a layer of low oxygen lead to a bottom layer of high oxygen? I do not think this manuscript is the place to discuss processes that lead to different distribution. If you still want to do so, please refer to appropriate literature.

Line 227: "weak" ? LINE 295 ok

Line 260: Why use absolute salinity? The convention is to use PSU for data and data products, and do the conversion to absolute salinity in a second step working with the data.
Yes, we just used absolute salinity and conservative temperature as a second step but the final data product contain the salinity in PSU and temperature

Table 5: What does the colors mean?
Added in figure 6 title :Color gradient: Blue to Green to Yellow indicate low to high values

Line 393: Hainbucher et al. do not consider any of the two cruises mentioned above
Changed

Table 2: Hainbucher was the chief scientist of the MSM72 cruise. Please double check the table. I would also keep only one name for chief scientist.
Updated

Line 803: Please be consistent in the language: I suggest using "adjustment" to what is applied to a cruise. So that table 3, it should be adjustments, assuming that those "suggested corrections" where actually applied.
See table 4 and the language of section 4 was updated

Line 838: Did you apply the adjustment? I suggest using the term adjustment to what is actually applied to the data product.
For each of the descriptions of the individual cruises, end by stating what adjustment was applied based on the evidence you provide.

Summary and conclusion: Why did you not mention the 1st QC here? Point 9 in the conclusion is not the conclusion based on this work and thus do not belong in that section (perhaps in the intro).
added

Line 1120: Strange sentence, please reformulate.

The information about two data products (one with and one without the adjustments, but both with the results of 1st QC) is an important piece of information,
and I think this belongs in the main manuscript, probably just before the conclusion section.
OK

Looking at the data page at PANGAEA, it is only one data set there. The text do not explain if this data set has the adjustments applied, or not. Please correct the text to clearly and explicitly describe the data set, rather than why and the context. Is this with adjustments, or not. And where is the other data set that the article states should be there?
YES PANGAEA IS UPDATED, under review

Also, and very importantly, what is the salinity value in the data product? The data description at PANGAEA states only salinity, but the paper states that you calculated absolute salinity, but never stated which of the absolute of practical salinity is used for the data product. This HAS to be explicitly stated. It is strongly recommended to use practical salinity for a data product. This also goes for table 4 in the paper.
Table 5